# Deep Learning for Comprehensive Analysis of Retinal Fundus Images: Detection of Systemic and Ocular Conditions

**DOI:** 10.3390/bioengineering12080840

**Published:** 2025-08-03

**Authors:** Mohammad Mahdi Aghabeigi Alooghareh, Mohammad Mohsen Sheikhey, Ali Sahafi, Habibollah Pirnejad, Amin Naemi

**Affiliations:** 1Department of Electrical and Computer Engineering, Lorestan University, Khorramabad 68151-44316, Iran; aghabeigi.mm@fe.lu.ac.ir (M.M.A.A.); sheikhey.m@lu.ac.ir (M.M.S.); 2Institute of Mechanical and Electrical Engineering, University of Southern Denmark, 5230 Odense, Denmark; 3Department of Family Medicine, Amsterdam University Medical Center, 1105 Amsterdam, The Netherlands; h.pirnejad@amsterdamumc.nl; 4Nordcee, Department of Biology, University of Southern Denmark, 5230 Odense, Denmark

**Keywords:** deep learning, retinal disease, fundus images, vision transformers, convolutional neural networks, explainable AI

## Abstract

The retina offers a unique window into both ocular and systemic health, motivating the development of AI-based tools for disease screening and risk assessment. In this study, we present a comprehensive evaluation of six state-of-the-art deep neural networks, including convolutional neural networks and vision transformer architectures, on the Brazilian Multilabel Ophthalmological Dataset (BRSET), comprising 16,266 fundus images annotated for multiple clinical and demographic labels. We explored seven classification tasks: *Diabetes*, *Diabetic Retinopathy (2-class)*, *Diabetic Retinopathy (3-class)*, *Hypertension*, *Hypertensive Retinopathy*, *Drusen*, and *Sex* classification. Models were evaluated using precision, recall, F1-score, accuracy, and AUC. Among all models, the Swin-L generally delivered the best performance across scenarios for *Diabetes* (AUC = 0.88, weighted F1-score = 0.86), *Diabetic Retinopathy (2-class)* (AUC = 0.98, weighted F1-score = 0.95), *Diabetic Retinopathy (3-class)* (macro AUC = 0.98, weighted F1-score = 0.95), *Hypertension* (AUC = 0.85, weighted F1-score = 0.79), *Hypertensive Retinopathy* (AUC = 0.81, weighted F1-score = 0.97), *Drusen* detection (AUC = 0.93, weighted F1-score = 0.90), and *Sex* classification (AUC = 0.87, weighted F1-score = 0.80). These results reflect excellent to outstanding diagnostic performance. We also employed gradient-based saliency maps to enhance explainability and visualize decision-relevant retinal features. Our findings underscore the potential of deep learning, particularly vision transformer models, to deliver accurate, interpretable, and clinically meaningful screening tools for retinal and systemic disease detection.

## 1. Introduction

Chronic retinal diseases such as Diabetic Retinopathy (DR), Hypertensive Retinopathy (HR), and Age-related Macular Degeneration (AMD) are leading causes of vision loss worldwide [1,2]. DR and HR are primarily caused by systemic conditions, diabetes mellitus and chronic hypertension, which lead to characteristic microvascular and vascular alterations in the retina. Current screening programs often fail to detect the disease until irreversible retinal damage has occurred [1,3,4]. Chronic hypertension also manifests in the retina, where elevated blood pressure induces characteristic vascular changes (hypertensive retinopathy) that not only threaten vision but serve as biomarkers of systemic cardiovascular risk. Continuous monitoring of these conditions is critical for early detection. Timely intervention (e.g., laser treatment in DR or injections for neovascular AMD) can prevent or delay blindness [5,6]. Similarly, AMD is a prevalent degenerative retinal disease and one of the leading causes of severe irreversible visual impairment in older populations. The number of people with AMD is projected to reach 288 million by 2040 [7], which reflects the impact of aging demographics. One of the key indicators in the early and intermediate stages of AMD is the presence of drusen, which is yellowish extracellular deposits that accumulate between the retinal pigment epithelium and Bruch’s membrane [8]. Drusen indicates early AMD and increases the risk of progression to neovascular (wet) AMD, as their presence is linked to retinal pigment epithelium dysfunction and inflammation [9].

Driven by advances in Artificial Intelligence (AI), the availability of large datasets, and increased computational power, deep learning has rapidly evolved from a research concept to a practical tool across many industries. In healthcare, AI applications are expanding quickly, with models increasingly matching or surpassing physician-level performance in various diagnostic tasks [10]. Specifically, Deep Neural Networks (DNNs) have demonstrated remarkable performance in medical image analysis, often matching or surpassing the expertise in specialized tasks [11,12,13]. One key application of AI in medical imaging is the detection and diagnosis of DR.

Gulshan et al. [14] developed a Convolutional Neural Network (CNN) for DR detection that achieved high sensitivity and specificity (>90%) in detecting referable DR from fundus images. The efficacy of the diagnosis was demonstrated in multiple cases of various diseases. For instance, Ting et al. [15] reported a deep learning system that could identify DR, glaucoma, and AMD with Area Under the Curve (AUC) values around 0.93–0.96 for various referable conditions. Notably, their algorithm detected referable AMD with 93% sensitivity and 89% specificity. Deep learning models have also been applied to specific lesion detection. For example, Keenan et al. [16] trained a CNN on 59,812 fundus images to automatically detect geographic atrophy, which is an advanced form of AMD. Recent advancements in deep learning have led to more accurate automated detection of drusen in retinal fundus images. This progress supports earlier diagnosis and better management of AMD [8,17,18,19]. For instance, Sadek et al. [19] utilized features extracted from pre-trained CNN to classify bright retinal lesions, such as drusen, and they achieved an accuracy between 91% and 92% across multiple datasets using color fundus images. In another study, a multi-scale deep learning model was developed for drusen segmentation in 57 high-resolution fundus images, and both global and local information were effectively leveraged to improve detection performance [18]. In 2018, the United States Food and Drug Administration (FDA) approved the first autonomous AI diagnostic system for diabetic retinopathy after a prospective trial in primary care by Abràmoff et al. [20]. The system achieved a sensitivity of 87% and a specificity of 91% for referable DR.

Beyond ophthalmic diseases, the retina is increasingly recognized as a window into systemic health, and AI methods leverage this to assess chronic disease risk factors. In a seminal study, Poplin et al. [21] employed deep learning on fundus images to predict cardiovascular risk factors that are not directly visible to clinicians. Their model could estimate attributes like age, gender, smoking status, blood pressure, and even the risk of major cardiac events from retinal images. For example, systolic blood pressure was predicted with a mean error of about 11 mmHg, and the algorithm identified patients who would experience a cardiac event within five years with an AUC of ∼0.70 [21]. The AI accomplished this by analyzing retinal microvascular features, such as vessel caliber and tortuosity that correlate with systemic vascular health. Moreover, researchers have shown that hypertensive retinal changes detectable by AI can serve as an early indicator for stroke and other complications of hypertension [1]. Recently, the scope of retinal AI has expanded to metabolic disorders. Lee et al. [22] developed a deep learning model based on vision transformers to classify metabolic syndrome using fundus photos. Their system achieved an AUC of 0.78 in distinguishing individuals with metabolic syndrome from healthy controls using only retinal images, and performance rose above 0.87 when combining image analysis with basic clinical data.

In line with the abovementioned findings, recent advances in AI show that diabetes can be predicted from retinal images, even without visible retinopathy. Subtle features like vessel caliber and texture carry predictive signals. Poplin et al. [21] used deep learning to predict diabetes and cardiovascular risks from fundus images. Lee et al. [22] achieved an AUC of 0.78 for metabolic syndrome detection using retinal images. These studies support the retina’s value as a non-invasive tool for early diabetes screening.

Building on CNN successes in retinal analysis, recent research is turning to Vision Transformers (ViTs) to boost diagnostic accuracy and efficiency. ViTs replace traditional convolution with a self-attention mechanism that models long-range relationships within an image. This global context modeling is advantageous for retinal images, which often contain multiple lesions or diffuse patterns that span across the fundus. ViTs have already shown promising performance in various classification tasks previously dominated by CNNs [23]. Recent studies suggest that ViT-based models can at least match CNN performance in retinal disease detection, and in some cases offer superior generalization. For example, Goh et al. [23] conducted a head-to-head comparison of ViTs and CNNs for detecting referable DR and found that transformer models achieved comparable sensitivity while slightly improving specificity, which highlights the viability of transformers for fundus screening. ViTs are especially appealing for multi-label or multi-disease classification problems, where multiple pathologies may co-exist in one image. Wang et al. [24] proposed a novel transformer-based architecture for simultaneous identification of multiple retinal conditions, including DR, glaucoma, and AMD. They also demonstrated improved performance over state-of-the-art CNNs on a public multi-disease fundus image dataset. Their transformer model outperformed established CNN architectures like ResNet, VGG, and DenseNet in multi-disease classification, which indicates that self-attention mechanisms can better capture the diverse features needed for comprehensive retinal assessment [24]. Zhou et al. [25] introduced RETFound, a self-supervised ViT model trained on 1.6 million unlabeled fundus images, and then fine-tuned it for various ocular disease tasks. RETFound achieved superior accuracy compared to CNNs across multiple external test sets. For instance, an AUC of 0.94 was achieved on a DR grading task, demonstrating superior performance compared to CNN baselines pre-trained on ImageNet [25].

In this study, we implement and comprehensively evaluate a total of six state-of-the-art DNNs, incorporating both ViT-based transformer architectures (PVTv2-B5, Twins-SVT-L, CSWin-B, Swin-L) and CNN backbones (RegNetY-16GF and SE-ResNeXt101), for multi-label classification of retinal diseases using the Brazilian Multilabel Ophthalmological Dataset (BRSET). The evaluation covers seven classification scenarios, namely, *Diabetes*, *Diabetic Retinopathy* (2-class and 3-class grading), *Hypertension*, *Hypertensive Retinopathy*, *Drusen*, and *Sex* classification. The performance of implemented models is evaluated using precision, recall, F1-score, accuracy, Receiver Operating Characteristic (ROC) curve, AUC, and Precision–Recall (PR) curve. Additionally, we employ gradient-based saliency maps to explain how the models interpret different regions of the images and make classification decisions.

## 2. Materials and Methods

Figure 1 provides an overview of the complete methodology adopted in this study. The process begins with acquisition of the BRSET dataset from PhysioNet, which comprises 16,266 labeled retinal fundus images. This dataset undergoes a preprocessing step that includes cleaning and validation in collaboration with both data scientists and ophthalmologists to ensure consistency and clinical reliability. Subsequently, seven clinically relevant classification scenarios are defined, including systemic conditions like *Diabetes* and *Hypertension*, as well as ocular findings such as *Diabetic Retinopathy*, *Hypertensive Retinopathy*, and *Drusen*. The dataset is split into training and testing subsets. Then, six state-of-the-art deep learning models, comprising both CNN and ViT architectures, including RegNetY-16GF, SE-ResNeXt101-32×8d, PVTv2-B5, Twins-SVT-L, CSWin-B, and Swin-L, are trained for multi-label classification. The performance of each model is assessed using standard metrics, such as accuracy, precision, recall, F1-score, ROC curve, AUC, and PR curve. Finally, to enhance clinical interpretability, three gradient-based approaches are employed to visualize the image regions most influential in the model’s decision-making process. This pipeline ensures both robustness and explainability in the development of AI-based tools for retinal image analysis.

### 2.1. Dataset and Preprocessing

The BRSET v1.0.1 dataset was utilized in this study. It contains 16,266 retinal fundus images from 8524 patients (mean age 57.6 years), primarily collected from three ophthalmology centers in São Paulo, Brazil, between 2010 and 2020 [26]. In terms of metadata quality, no duplicate records were found. The missing values were limited to three columns: patient age (33.47%), duration of diabetes (88.26%), and insulin use (89.46%), the latter two only for diabetic patients. Table 1 summarizes the number of images associated with each target label investigated in this study.

The dataset was curated to facilitate AI research focused on retinal diseases and other conditions detectable through retinal fundus images, such as diabetes and hypertension. This dataset is multi-labeled for various retinal pathologies, including DR graded according to the International Clinical Diabetic Retinopathy (ICDR) and Scottish Diabetic Retinopathy Grading (SDRG) systems, HR, AMD, drusen, increased Cup-to-Disc Ratio (CDR), and other retinal anomalies. Annotations cover disease-specific and multifactorial conditions and image quality metrics such as focus and illumination. Hypertension cases were inferred from comorbidity data rather than direct labeling. This dataset provides a diverse and underrepresented population for developing and benchmarking AI models for fundus images [2,26].

### 2.2. Scenarios

In this study, we utilized seven scenarios based on our discussion with an expert ophthalmologist. In this section, we present a short description about these scenarios and their importance.

*Diabetes Classification*: The goal of this scenario is to distinguish between retinal images of individuals with and without diabetes. This scenario is important because it demonstrates the potential of retinal fundus images to serve as non-invasive biomarkers for systemic diseases like diabetes. Accurate classification of diabetic patients can aid in opportunistic screening and early identification of at-risk individuals, especially in populations with limited access to endocrinological testing or electronic health records [21].

*Diabetic Retinopathy (2-Class) Classification*: The objective of this binary classification task is to detect the presence or absence of DR in retinal fundus photographs. This scenario is clinically significant as it addresses the early detection of DR, which is a major complication of diabetes and a leading cause of vision loss globally [14]. Early identification of DR can guide timely interventions and reduce the risk of blindness.

*Diabetic Retinopathy (3-Class) Classification*: The ICDR severity scale categorizes DR into five stages, including no apparent retinopathy, mild, moderate, severe non-proliferative DR (NPDR), and proliferative DR (PDR) [27]. For analytical purposes, these stages are often consolidated into three clinically relevant categories: normal (no apparent retinopathy), NPDR (encompassing mild, moderate, and severe NPDR), and PDR, which streamlines analysis and interpretation in both clinical and research settings [28]. This scenario is critical because it mirrors real-world diagnostic needs where differentiating between disease stages determines follow-up intervals and therapeutic approaches.

*Hypertension Classification*: The purpose of this task is to predict whether a patient has systemic hypertension based on the characteristics of the retinal imaging. It is an important task because hypertensive changes in the retina reflect broader cardiovascular risks and are often underdiagnosed [29].

*Hypertensive Retinopathy Classification*: This scenario seeks to identify HR, which presents as retinal signs of high blood pressure, such as arteriolar narrowing, hemorrhages, or cotton wool spots [29]. This task is critical because it targets end-organ retinal damage from hypertension. Detecting these lesions aids both ocular and systemic disease management.

*Drusen Classification*: This classification task aims to detect the presence of drusen, extracellular deposits that are key early indicators of AMD. This scenario is critical to initiate prompt surveillance and preventive measures in patients at risk of AMD progression [30]. AI-powered detection of Drusen, significantly improves the early diagnosis of age-related vision loss.

*Sex Classification*: This scenario involves classifying patient sex based solely on retinal fundus images. It is not a disease-specific task, but it serves to explore the presence of sex-linked anatomical and risk factor differences in retinal features. Understanding these differences can be important for developing fair and unbiased AI systems and may also uncover physiological patterns that influence disease presentation and model performance [21].

In each binary classification scenario, the label “0” corresponds to the non-disease class, and “1” indicates the presence of disease. For a more comprehensive evaluation, both macro-averaged and weighted-averaged values are reported for each metric. Additionally, in the *Diabetic Retinopathy (3-class)* scenario, the classes “0”, “1”, and “2” represent normal, NPDR, and PDR stages, respectively [28]. For the *Sex* classification task, “0” denotes female and “1” represents male.

### 2.3. Models Development

We split the dataset on a patient-wise basis into training (75%) and testing (25%) sets and developed six state-of-the-art DNNs. Brief descriptions of these models are presented in Table 2. We evaluated a diverse set of CNN and ViT architectures on the BRSET dataset and selected six backbone models for detailed analysis. These include Regular Network Y with an approximately 16 Giga Floating Point Operations (GFLOPs) network (RegNetY-16GF), Squeeze-and-Excitation (SE-ResNeXt101-32×8d), Pyramid Vision Transformer v2–Base 5 (PVTv2-B5), Twins Spatially Separable Vision Transformer Large (Twins-SVT-L), Cross-Shaped Window Transformer Base (CSWin-B), and Swin Transformer Large (Swin-L). CNNs use shared, learned filters to hierarchically extract local patterns, such as edges and textures, to achieve strong data efficiency and translation equivariance. In contrast, ViTs divide images into patches and apply global self-attention mechanisms to capture long-range dependencies. However, this approach incurs a quadratic computational cost and typically requires larger amounts of training data. Many ViT variants mitigate this trade-off by introducing hierarchical or windowed attention [31].

SE-ResNeXt101-32×8d enhances channel-wise feature representations by employing Squeeze and Excitation (SE) modules that use global average pooling followed by a lightweight gating mechanism to recalibrate channel responses [32]. With 93.6 million parameters, it achieves competitive ImageNet performance while maintaining CNN-level efficiency, although its attention is limited to channel reweighting. In addition, RegNetY-16GF adopts a principled design space of repeated 3 × 3 convolutions, parameterized by linear rules for width, depth, and group size, and integrates SE blocks for channel attention. With around 83.6 million parameters and 15.9 GFLOPs, it scales smoothly and often outperforms ResNets of similar size. However, it still depends on depth rather than explicit spatial attention to the long-range context of the model [33].

Among ViTs, Swin-L builds a hierarchical feature pyramid with shifted-window self-attention, achieving linear complexity and a large receptive field, but limited cross-window interaction [34]. Twins transformer achieves performance comparable to the Swin Transformer through a dual-attention design while using fewer computational resources (99 million parameters) [35]. CSWin Transformer utilizes self-attention in cross-shaped windows and locally enhanced positional encodings to efficiently capture the near-global context. It achieves performance comparable to Swin and Twins Transformers while using fewer parameters, albeit with increased architectural complexity [36]. Finally, PVTv2-B5 integrates linear attention mechanisms, overlapping patch embeddings, and convolutional feedforward networks within a pyramid architecture, achieving state-of-the-art performance on ImageNet-1K with reduced computational complexity compared to pure ViTs [37].

**Table 2 bioengineering-12-00840-t002:** Overview of the models’ characteristics implemented in this study.

Model	Architecture Type	Key Features	FLOPs (Billion)	Params (Million)
**RegNetY-16GF [33]**	CNN	Simple stage-wise design (width/depth design space) + SE channel blocks	15.9	83.6
**SE-ResNeXt101-32×8d [32]**	CNN	Incorporates SE blocks for channel attention; improved feature recalibration	16.5	93.6
**PVTv2-B5 [37]**	Pyramid ViT	Linear-attention (reduced tokens); overlapping patch embedding; conv-FFN	11.8	82
**Twins-SVT-L [35]**	Pyramid ViT	Separable local/global attention (LSA + GSA); positional encoding	15.1	99.2
**CSWin-B [36]**	Hierarchical ViT	Cross-shaped window attention (horizontal + vertical stripes); Locally-enhanced PE	15	78
**Swin-L [34]**	Hierarchical ViT	Shifted local window self-attention; multi-stage/pyramid	34.5	197

Table 3 presents the hyperparameters used for training of all models in this study. All input images were resized to 224×224 pixels. During training, a batch size of 16 was used. The models were trained for a maximum of 50 epochs using the Adam optimizer with an initial learning rate of 1 × 10^−5^, which remained fixed throughout the training process [38]. Early stopping with a patience of 7 epochs was applied to prevent overfitting. To address class imbalance, a weighted cross-entropy loss function was used. Data augmentation techniques, including random cropping, horizontal flipping, and random rotation, were applied to improve generalization [39]. For CNN-based models, ReLU was used as the activation function, while GELU was used for ViT architectures [40,41]. All models were initialized with ImageNet-1K pre-trained weights and full fine-tuning was performed, updating all model parameters [42]. Based on the BRSET dataset, the final classification head was replaced according to the classification task (binary or three-class). The experiments were implemented in PyTorch (version 2.6.0) and a fixed random seed of 42 was set to ensure reproducibility.

### 2.4. Performance Metrics

In evaluating the performance of classification models, particularly in medical image analysis, various metrics are employed to capture different aspects of model behavior. Precision measures the proportion of correctly identified positive cases out of all predicted positives. Recall or sensitivity quantifies the proportion of actual positive cases correctly identified, and specificity measures the proportion of correctly identified negative cases out of all actual negative cases. These are defined as:(1)$$\mathrm{Precision}=\frac{\mathrm{TP}}{\mathrm{TP}+\mathrm{FP}}\;\;,\;\;\mathrm{Recall}=\frac{\mathrm{TP}}{\mathrm{TP}+\mathrm{FN}}\;\;,\;\;\mathrm{Specificity}=\frac{\mathrm{TN}}{\mathrm{TN}+\mathrm{FP}}$$
where TP is the number of true positives, FP is false positives, and FN is false negatives.

To balance the trade-off between precision and recall, the F1-score is used as the harmonic mean of the two:(2)$$F1-\mathrm{score}=2\times \frac{\mathrm{Precision}\times \mathrm{Recall}}{\mathrm{Precision}+\mathrm{Recall}}$$

Accuracy is another commonly used metric that reflects the overall proportion of correct predictions out of all predictions.(3)$$\mathrm{Accuracy}=\frac{\mathrm{TP}+\mathrm{TN}}{\mathrm{TP}+\mathrm{TN}+\mathrm{FP}+\mathrm{FN}}$$

To evaluate classifier performance across all decision thresholds, the ROC curve is used. It plots the True Positive Rate versus the False Positive Rate. The AUC summarizes the ROC curve into a single value, where a value of 1 indicates perfect classification and 0.5 suggests random guessing. The True Positive Rate, also known as sensitivity or recall, and the False Positive Rate, equal to 1-Specificity, are defined as:(4)$$\mathrm{True}\;\mathrm{Positive}\;\mathrm{Rate}=\frac{\mathrm{TP}}{\mathrm{TP}+\mathrm{FN}}\;\;,\;\;\mathrm{False}\;\mathrm{Positive}\;\mathrm{Rate}=\frac{\mathrm{FP}}{\mathrm{FP}+\mathrm{TN}}$$

Together, these metrics provide a comprehensive view of model performance, especially in multi-label and imbalanced classification settings common in medical imaging. Moreover, to have a better estimation about the performance of AI models, we utilized 5-fold cross-validation to calculate the results.

### 2.5. Explainability

To enhance model interpretability and gain insights into the decision-making process of our deep learning classifiers, we employed three gradient-based saliency map techniques named Vanilla Gradients [43], Integrated Gradients [44], and GradientSHAP [45]. This method computes the gradient of the predicted class score with respect to each input pixel, effectively highlighting the regions of the retinal fundus image that most strongly influence the model’s prediction. It is worth mentioning that GradientSHAP is an established variant of SHapley Additive exPlanations (SHAP) [46], adapted to work with ViT-based architectures through Captum’s implementation.

By visualizing these gradients, we are able to qualitatively assess whether the model is focusing on clinically relevant structures, such as microaneurysms, drusen deposits, or vascular changes.

## 3. Results

In this study, we defined seven classification scenarios and evaluated the performance of six state-of-the-art DNNs on the BRSET dataset. A detailed comparison of the models across all disease categories based on precision, recall, F1-score, accuracy, and AUC is presented in Table 4.

The ROC curve assesses a binary classifier through a plot of sensitivity (True Positive Rate) against 1-specificity (False Positive Rate) across all thresholds. In multi-class tasks, the one-versus-rest approach forms a separate ROC for each class, marks that class positive and the others negative, and the resultant AUC shows how well the model detects that disease. Figure 2 illustrates the ROC curves for six different binary classification tasks across all evaluated models. As expected, higher AUC values correspond to better classification performance. For the tasks of *Diabetes*, *Diabetic Retinopathy (2-class)*, *Hypertension*, *Drusen*, and *Sex* (Figure 2a–c,e,f), all models demonstrate competitive and closely aligned AUCs. However, ViT-based architectures, such as Swin-L and Twins-SVT-L, slightly outperform their CNN-based counterparts. In contrast, for the *Hypertensive Retinopathy* task (Figure 2d), ViTs show noticeably superior AUC values compared to CNN models. This performance gap may reflect the enhanced capability of ViTs to learn from smaller or imbalanced datasets, leveraging their self-attention mechanisms [47]. Notably, Swin-L consistently achieves the best or near-best performance in most scenarios. This can be attributed to its hierarchical feature representation and shifted window attention mechanism, which effectively capture both local and global contextual information [34].

Figure 3 presents the class-wise ROC curves for the *Diabetic Retinopathy (3-Class)* using a one-vs-rest strategy. For each model, ROC curves are plotted for the three classes: class 0 (Normal), class 1 (NPDR), and class 2 (PDR), as shown in Figure 3a–f. All models except RegNetY-16GF demonstrate promising AUCs in distinguishing the DR classes. Furthermore, a clear performance advantage of ViTs (Figure 3c–f) over CNN-based models (Figure 3a,b) is evident. This underscores the superior ability of ViTs to extract global contextual features from retinal fundus images, which proves particularly advantageous in complex, multi-class clinical classification scenarios.

In this study, we prioritized the AUC as the primary evaluation metric due to its robustness in handling class imbalance and its ability to reflect the model’s performance across all classification thresholds. Our results show promising diagnostic performance across multiple retinal disease classification scenarios using state-of-the-art DNNs. According to the AUC interpretive framework proposed by Mandrekar et al. [48], wherein an AUC > 0.9 is considered *outstanding*, 0.8–0.9 as *excellent*, and 0.7–0.8 as *acceptable*, the majority of our model predictions fall into the *excellent* to *outstanding* categories. For instance, as seen in Table 4, for the *Diabetic Retinopathy (2-Class)* classification task, all models achieved AUC values between 0.95 and 0.98, which clearly fall in the *outstanding* category. Similarly, the *Drusen* classification scenario achieved AUCs ranging from 0.88 to 0.93 across models, placing them in the *excellent* to *outstanding* range.

In the *Diabetes*, *Hypertension*, and *Sex* classification scenarios, AUC values predominantly ranged from 0.80 to 0.88, corresponding to *excellent* diagnostic performance. Notably, in the *Diabetic Retinopathy (3-Class)* scenario, class-specific AUCs reached up to 0.99 for proliferative stages, reflecting *outstanding* model capability in differentiating disease severity. The *Hypertensive Retinopathy* task demonstrated more moderate performance, with AUCs between 0.67 and 0.82, spanning the *acceptable* to *excellent* categories. This relatively lower performance likely reflects class imbalance and may also be influenced by subtle or overlapping imaging features. Overall, these findings indicate that the evaluated models can deliver clinically meaningful predictions across most disease contexts, particularly in tasks where disease markers are visually distinct.

Figure 4 presents the PR curves for six binary classification scenarios for (a) *Diabetes*, (b) *Diabetic Retinopathy (2-class)*, (c) *Hypertension*, (d) *Hypertensive Retinopathy*, (e) *Drusen*, and (f) *Sex* classification. These plots provide a detailed view of model performance under varying classification thresholds. In all scenarios, particularly in clinically challenging tasks like Drusen (Figure 4e), ViT-based models, especially Swin-L and CSWin-B, consistently demonstrate superior PR trade-offs. Notably, in the low-prevalence scenario of Hypertensive Retinopathy (Figure 4d), all models exhibit markedly lower APs, but ViT architectures still outperform CNN baselines. These results underscore the ability of advanced ViTs to maintain high precision even at high recall levels, which is critical for minimizing false positives in clinical settings.

Figure 5 displays the one-vs-rest PR curves for the *Diabetic Retinopathy (3-class)* classification scenario. Each subplot shows PR curves for class 0 (Normal), class 1 (NPDR), and class 2 (PDR). Among the models, Swin-L (Figure 5f) and CSWin-B (Figure 5e) achieve the highest average precision for the more complex PDR class (AP = 0.87), reflecting their ability to differentiate severe disease stages. In contrast, CNN-based models like RegNetY-16GF (Figure 5a) show substantially lower APs for class 1 and 2. Overall, these PR curves highlight the advantage of ViT models in handling nuanced, multi-class classification tasks, particularly in scenarios where inter-class boundaries are subtle and clinically critical.

Another critical aspect in developing and deploying AI models in clinical practice is explainability and transparency. According to current best practices in medical AI, transparency is essential to foster clinician trust and identify biases and ensure the safe and ethical use of automated systems [49]. As shown in Figure 6, we implemented three explainability techniques, namely, Integrated Gradients, Vanilla Gradient, and GradientSHAP to enhance model interpretability and analyze which retinal features most influenced the Swin-L model’s predictions across all classification scenarios. While Integrated Gradients and Vanilla Gradient are well-established saliency methods, SHAP was originally developed for models with well-defined feature semantics, such as tree ensembles or tabular neural networks. Therefore, direct application to ViT, including Swin-L, is not entirely straightforward. To address this, we adapted Captum’s implementation of GradientSHAP to generate pixel-level attributions as a practical approximation of SHAP values for ViTs [45]. Though these do not represent true SHAP explanations in the context of patch-based token representations, they nonetheless offer a reliable and interpretable distribution of saliency across the input image.

Across all three methods, the visualizations in Figure 6 consistently highlight clinically relevant retinal structures. For instance, in *Diabetic Retinopathy (2-class)* (Figure 6b) and *Hypertensive Retinopathy* (Figure 6e), all saliency techniques focus on vascular arcades, aligning with clinical landmarks for these conditions. In contrast, *Drusen* detection (Figure 6f) reveals attention in the macular region, while *Sex* classification (Figure 6g) emphasizes the optic disc area. The saliency maps produced by GradientSHAP closely resemble those generated by Integrated Gradients and Vanilla Gradient that confirms their reliability. Despite inherent architectural differences between ViTs and traditional CNNs, these interpretability tools serve as a vital bridge between AI models and clinical trust that explain the model’s decision-making process and support clinical trust in AI-assisted diagnostics [50,51,52,53].

The average inference and saliency map generation times per image for the six DNNs were evaluated and are presented in Table 5. All experiments were performed on an NVIDIA T4 GPU using 224 × 224 resolution images. Among the models, RegNetY-16GF demonstrated the fastest execution time, with inference taking only 13.80 ms (batch size 1) and saliency map generation 50.95 ms. It shows its suitability for time-sensitive clinical applications. In contrast, CSWin-B had the highest computational demand, with inference time of 68.56 ms and saliency map generation taking 204.82 ms, reflecting its greater architectural complexity. For a batch size of 16, all models showed improved inference speeds due to hardware parallelism, with RegNetY-16GF again being the fastest at 9.60 ms. These results highlight important trade-offs between model complexity and runtime efficiency, which are crucial considerations for deployment in real-world AI-assisted screening systems. While both Swin-L and Twins-SVT-L achieved near-identical diagnostic performance (about 1% relative advantage in AUC for Swin-L across most scenarios), Twins-SVT-L demonstrated significantly superior computational efficiency attributable to its dual-attention design [35]. At batch size 1, Twins-SVT-L reduced inference time by 38.8% and saliency map generation by 23.1% compared to Swin-L. This efficiency gap widened at batch size 16, where Twins-SVT-L processed images 56% faster. These features deliver a substantial speed advantage with only a marginal diagnostic difference that makes Twins-SVT-L a better choice for real-time clinical deployment.

## 4. Discussion

This study demonstrates the efficacy of state-of-the-art DNNs, particularly, ViT-based architectures, in classifying a range of systemic and ocular conditions from retinal fundus images. Among the six models evaluated, Swin-L consistently performed at or near the top across all seven classification scenarios. It indicates the strength of hierarchical transformer-based feature extraction in retinal imaging tasks. Our findings reinforce the retina’s utility not only for ocular diagnostics but also as a window into broader systemic health conditions such as hypertension and diabetes.

The diagnostic performance observed across the seven classification scenarios further underscores the strength of our approach. Based on Mandrekar el al.’s [48] categorization, most classification outcomes in this study fall into the top tiers. Specifically, Swin-L achieved *outstanding* AUC values in tasks such as *Diabetic Retinopathy (2-Class)* (0.98) and *Drusen* detection (0.93), and *excellent* performance in the *Diabetes* scenario (0.88), *Hypertension* task (0.85), and *Sex* classification (0.87). Even in more nuanced tasks like the *Hypertensive Retinopathy* task, where subtle visual cues challenge detection, Swin-L reached an AUC of 0.81, still within the *excellent* range. These consistent results across diverse disease types and task complexities affirm the robustness, clinical relevance, and translational potential of our transformer-based approach when applied to fundus image analysis.

To establish a robust benchmark, we compared our best-performing model, Swin-L, with other evaluated models on the BRSET dataset. As shown in Table 6, our study provides the most comprehensive evaluation to date, reporting per-class, macro-, and weighted-averaged values across several classification tasks. The top-performing results are highlighted for clarity. Swin-L consistently delivered the strongest performance based on the two most informative metrics, AUC and F1-score, reflecting both discrimination ability and the balance between precision and recall. It achieved the highest AUC across all tasks except *Sex* classification and led in macro F1-score for *Diabetes*, *Hypertension*, *Hypertensive Retinopathy*, and *Drusen* detection.

In addition, Swin-L showed the highest recall in all tasks, which reflects high sensitivity and a low number of false negatives, an important aspect of clinical screening. Precision was also highest in all scenarios except *Diabetic Retinopathy (2-Class)*, *Diabetic Retinopathy (3-Class)*, and *Sex* classification, where ConvNeXtV2-Large slightly outperformed it. Task-specific results further confirm Swin-L’s advantage. In *Diabetic Retinopathy (2-Class)*, it reached an AUC of 0.98 and a weighted F1-score of 0.84. For *Hypertensive Retinopathy*, it achieved a macro F1-score of 0.60 (versus 0.08) and AUC of 0.81 (versus 0.76) compared to ResNet-50 [54]. In *Drusen* detection, Swin-L attained an AUC of 0.93 and macro F1-score of 0.83.

In addition, this is the first study to explore *Hypertension* identification from retinal images using DNNs. Swin-L set a strong baseline (AUC = 0.85, macro F1 = 0.71) for future work in this area.

Despite the promising results, several challenges remain in integrating DNNs into clinical practice. One of the key barriers to the adoption of AI in real-world clinical settings is its “black-box” nature, which makes the decision-making process unclear and difficult for clinicians and patients to interpret or trust [49]. In other words, DNNs often lack transparency, as their predictions are derived from millions of parameters without clear reasoning that aligns with clinical logic. This lack of interpretability not only limits clinical confidence but also raises ethical and legal concerns, potentially increasing distrust in AI systems. To address this issue, our study incorporates explainable AI techniques, specifically, saliency maps generated using the gradient method [28,43] to visualize the regions of retinal fundus images that most influence the model’s predictions. These visual explanations can help bridge the gap between AI predictions and clinical understanding, offering clinicians greater insight into the model’s rationale. By highlighting disease-relevant features such as microaneurysms, vascular abnormalities, and drusen, saliency maps provide intuitive and transparent feedback. This added interpretability fosters trust among clinicians and patients alike, making AI-based tools more acceptable, safe, and ethically grounded for use in medical diagnostics.

While this study demonstrates several strengths, it has some limitations. First, even though the BRSET dataset is clinically rich, it is geographically limited to ophthalmology centers in Brazil. It may restrict the generalizability of the models to other populations with different ethnic, demographic, or disease prevalence profiles. Additionally, our study focused solely on static fundus photographs, but incorporating longitudinal imaging data or multimodal clinical inputs could further enhance diagnostic accuracy and clinical relevance.

## 5. Conclusions

This study demonstrates the feasibility and effectiveness of DNNs, particularly ViT architectures, for multi-label classification of retinal fundus images using the BRSET dataset. By evaluating six state-of-the-art models across seven clinically meaningful scenarios, including *Diabetes*, *Diabetic Retinopathy (2-class)* and *Diabetic Retinopathy (3-class)*, *Hypertension*, *Hypertensive Retinopathy*, *Drusen*, and *Sex* classification, we showed that most models achieved *excellent* to *outstanding* performance.

Among all models, the Swin-L model generally achieved the strongest performance across tasks. Quantitatively, Swin-L attained an AUC of 0.98 and weighted F1-score of 0.95 in the *Diabetic Retinopathy (2-class)* task, while for *Diabetic Retinopathy (3-class)*, it achieved a macro AUC of 0.98 and weighted F1-score of 0.95. In the detection of *Drusen*, a key indicator of age-related macular degeneration, Swin-L yielded an AUC of 0.93 and weighted F1-score of 0.90.

For systemic condition prediction, the model achieved an AUC of 0.88 and weighted F1-score of 0.86 for *Diabetes* classification, and an AUC of 0.85 with a weighted F1-score of 0.79 for the *Hypertension* scenario. Although *Hypertensive Retinopathy* posed a more challenging task due to subtle visual cues and class imbalance, Swin-L still reached an AUC of 0.81 and a weighted F1-score of 0.97. In the *Sex* classification scenario, it performed reliably with an AUC of 0.87 and weighted F1-score of 0.80, suggesting the presence of sex-linked anatomical patterns in retinal images.

These findings indicate that ViT-based DNNs can deliver *excellent* to *outstanding* diagnostic performance across a diverse set of clinical tasks. The integration of explainable AI techniques further enhanced model interpretability, revealing disease-relevant retinal structures and supporting transparent decision-making. Together, these results underscore the potential of AI to support non-invasive screening and early detection of both ocular and systemic diseases. Future work should address external validation and integration with multimodal data to further advance the clinical utility of these tools.

## Figures and Tables

**Figure 1 bioengineering-12-00840-f001:**
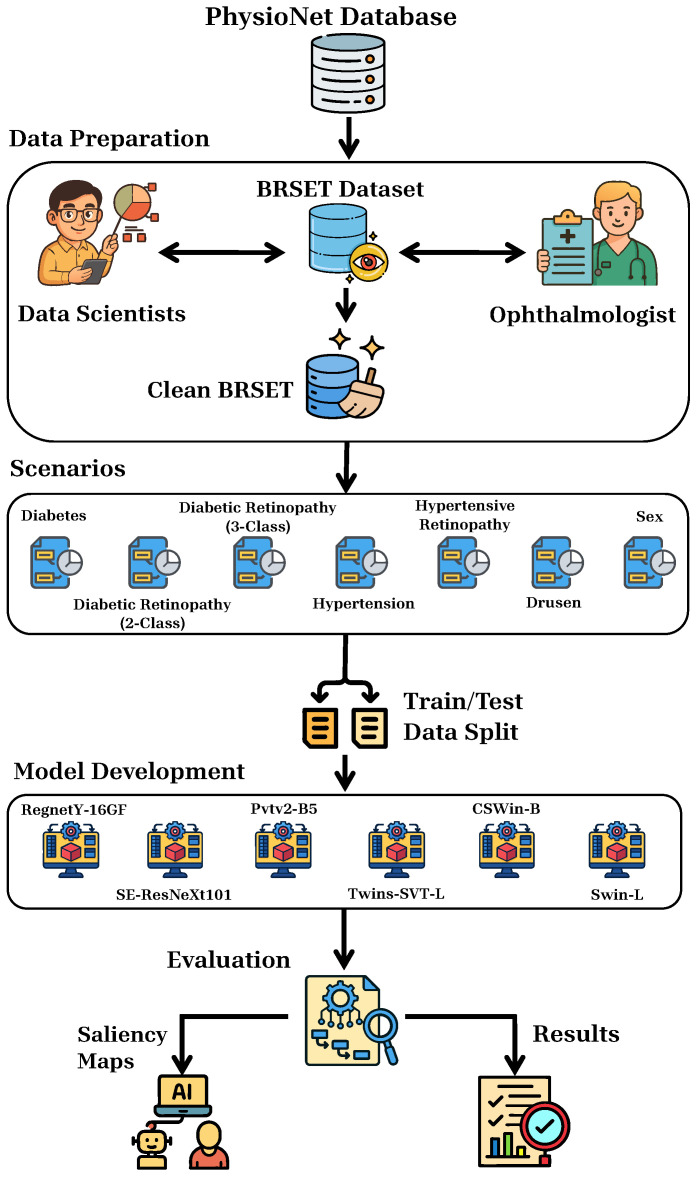
Methodology diagram of this study.

**Figure 2 bioengineering-12-00840-f002:**
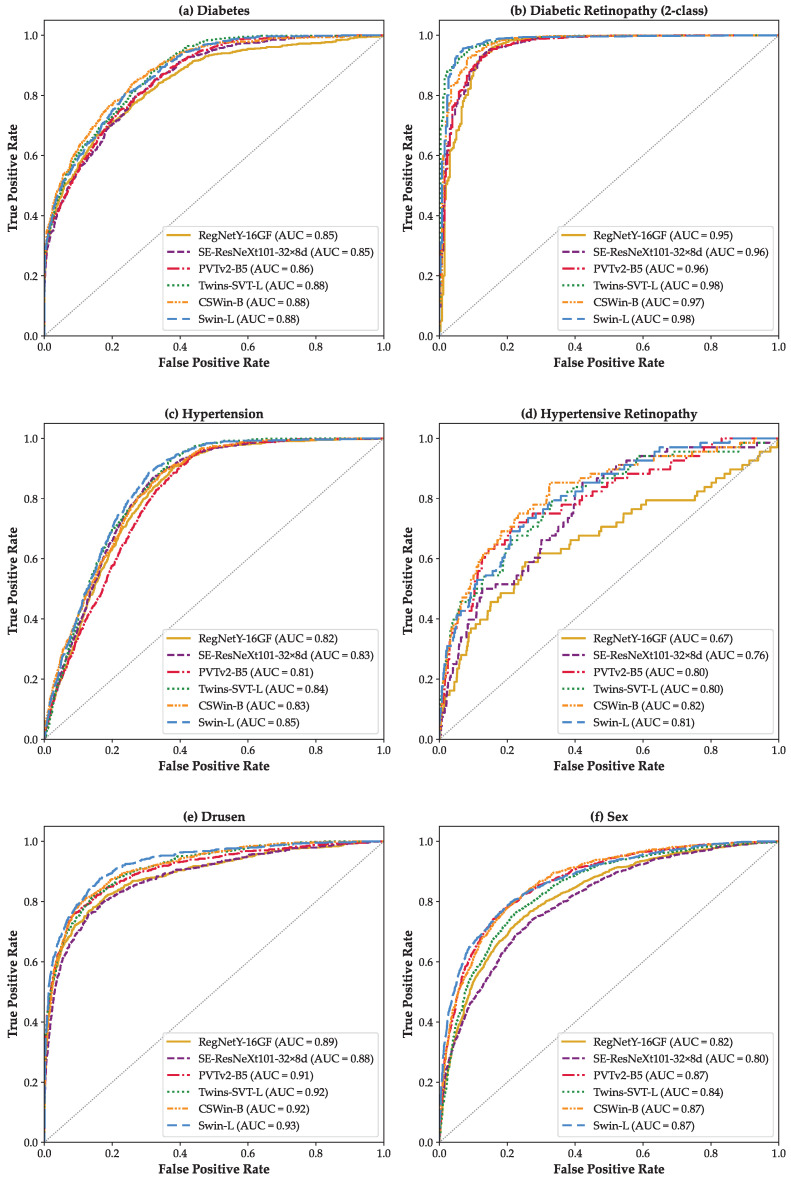
ROC curves for six binary scenarios: (**a**) *Diabetes*, (**b**) *Diabetic Retinopathy (2-class)*, (**c**) *Hypertension*, (**d**) *Hypertensive Retinopathy*, (**e**) *Drusen*, and (**f**) *Sex*.

**Figure 3 bioengineering-12-00840-f003:**
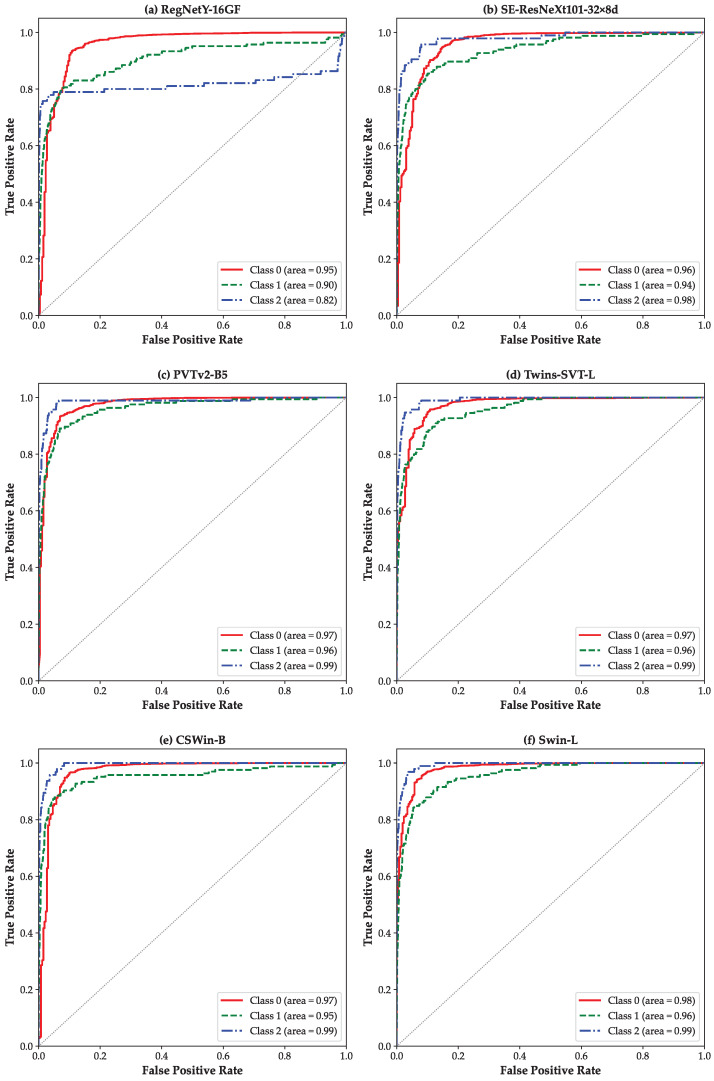
ROC curves based on the one-vs-rest strategy for the *Diabetic Retinopathy (3-class)* scenario: (**a**) RegNetY-16GF, (**b**) SE-ResNeXt101-32×8d, (**c**) PVTv2-B5, (**d**) Twins-SVT-L, (**e**) CSWin-B, and (**f**) Swin-L.

**Figure 4 bioengineering-12-00840-f004:**
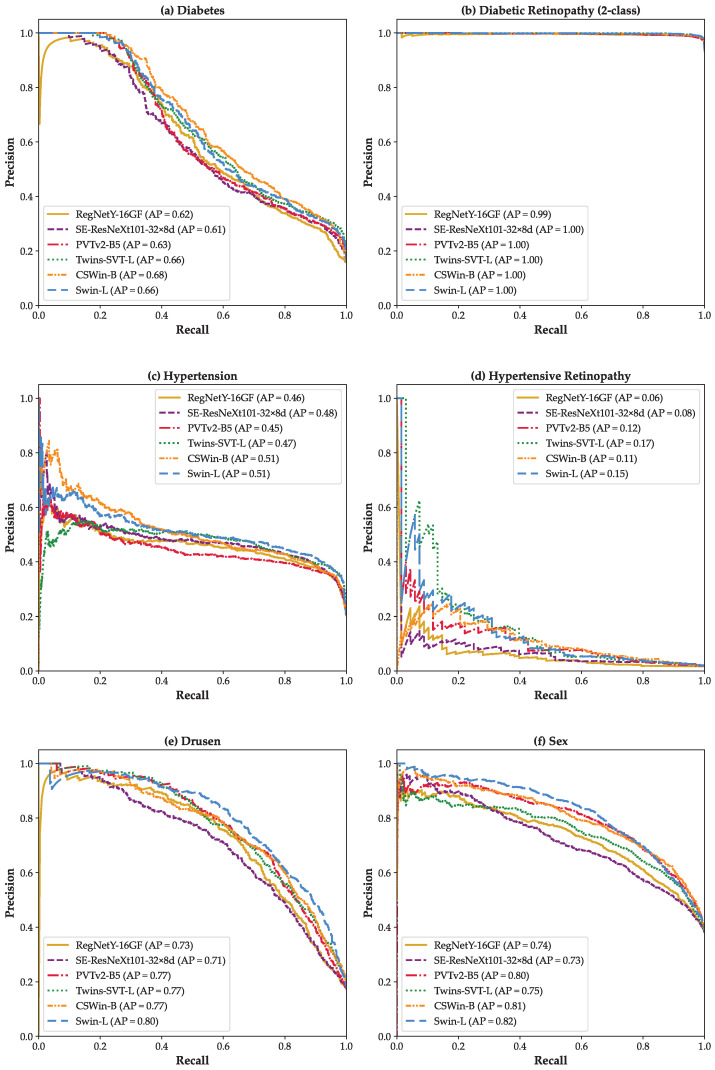
Precision–Recall curves for six binary scenarios: (**a**) *Diabetes*, (**b**) *Diabetic Retinopathy (2-class)*, (**c**) *Hypertension*, (**d**) *Hypertensive Retinopathy*, (**e**) *Drusen*, and (**f**) *Sex*.

**Figure 5 bioengineering-12-00840-f005:**
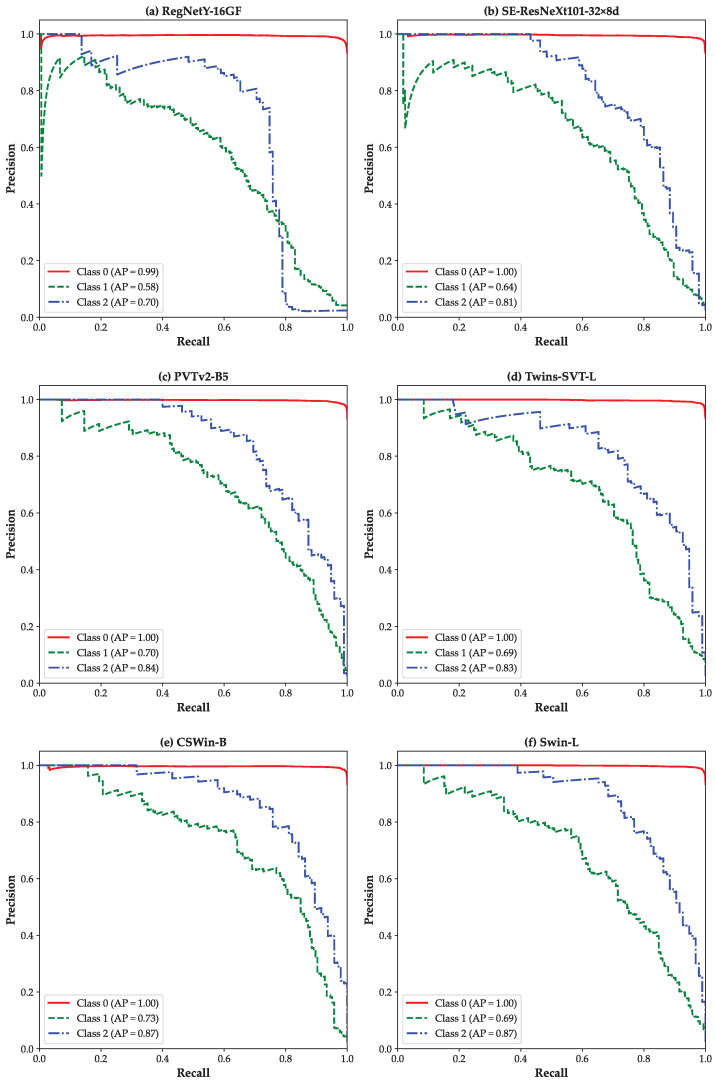
Precision–Recall curves based on the one-vs-rest strategy for the *Diabetic Retinopathy (3-class)* scenario: (**a**) RegNetY-16GF, (**b**) SE-ResNeXt101-32×8d, (**c**) PVTv2-B5, (**d**) Twins-SVT-L, (**e**) CSWin-B, and (**f**) Swin-L.

**Figure 6 bioengineering-12-00840-f006:**
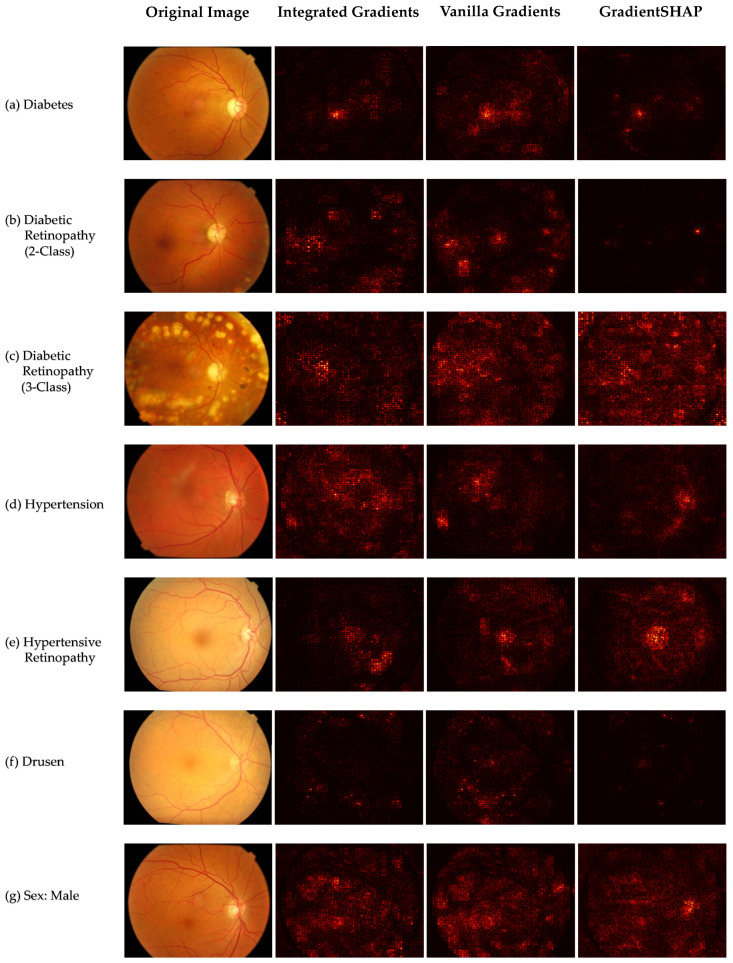
Original retinal fundus images and corresponding saliency maps (Integrated Gradients, Vanilla Gradient, and GradientSHAP) for Swin-L, our best model, for seven classification tasks: (**a**) *Diabetes*, (**b**) *Diabetic Retinopathy (2-class)*, (**c**) *Diabetic Retinopathy (3-class)*, (**d**) *Hypertension*, (**e**) *Hypertensive Retinopathy*, (**f**) *Drusen*, and (**g**) *Sex (Male)*.

**Table 1 bioengineering-12-00840-t001:** BRSET v1.0.1 dataset description.

Label	Image Numbers	Percentage (%)
**Normal**	11,001	67.63
**Diabetes**	2579	15.85
**Diabetic Retinopathy**	1070	65.80
**Hypertension**	3320	20.41
**Hypertensive Retinopathy**	284	1.75
**Drusen**	2833	17.42
**AMD**	299	1.84
**Sex**	10,052 (Female)	61.80

**Table 3 bioengineering-12-00840-t003:** Hyperparameters of all implemented models in this study.

Parameter	Value
Input Image Size	224×224
Batch Size	16
Learning Rate	1 × 10^−5^
Optimizer	Adam
Maximum Epochs	50
Early Stopping	Enabled (patience = 7)
Loss Function	Weighted Cross-Entropy Loss (with class weights)
Weight Decay	Not used
Data Augmentation	Random Crop, Horizontal Flip, Rotation
Activation Function	ReLU (CNNs)/GELU (ViTs)
Fine-Tuning	Full fine-tuning (all layers trained)
Pre-trained Weights	ImageNet-1K
Learning Rate Scheduler	None (fixed learning rate)
Random Seed	42
Framework	PyTorch v2.6.0

**Table 4 bioengineering-12-00840-t004:** Performance of six DNNs across seven classification scenarios.

Label	Model	Class	Precision	Recall	F1	Accuracy	AUC
**Diabetes**	RegNetY-16GF	0	0.92	0.90	0.91	0.85	0.85
		1	0.52	0.57	0.54		
		macro	0.72	0.74	0.73		
		weighted	0.85	0.85	0.85		
	SE-ResNeXt101 -32×8d	0	0.93	0.82	0.87	0.80	0.85
		1	0.41	0.69	0.52		
		macro	0.67	0.75	0.69		
		weighted	0.85	0.80	0.81		
	PVTv2-B5	0	0.93	0.84	0.88	0.81	0.86
		1	0.44	0.66	0.53		
		macro	0.68	0.75	0.70		
		weighted	0.85	0.81	0.83		
	Twins-SVT-L	0	0.93	0.89	0.91	0.85	0.88
		1	0.51	0.63	0.56		
		macro	0.72	0.76	0.73		
		weighted	0.86	0.85	0.85		
	CSWin-B	0	0.94	0.86	0.89	0.83	0.88
		1	0.47	0.69	0.56		
		macro	0.71	0.77	0.73		
		weighted	0.86	0.83	0.84		
	Swin-L	0	0.92	0.92	0.92	0.86	0.88
		1	0.57	0.57	0.57		
		macro	0.74	0.74	0.74		
		weighted	0.86	0.86	0.86		
**Diabetic** **Retinopathy** **(2-class)**	RegNetY-16GF	0	0.99	0.97	0.98	0.96	0.95
		1	0.65	0.82	0.73		
		macro	0.82	0.90	0.85		
		weighted	0.96	0.96	0.96		
	SE-ResNeXt101 -32×8d	0	0.99	0.96	0.97	0.95	0.96
		1	0.61	0.80	0.69		
		macro	0.80	0.88	0.83		
		weighted	0.96	0.95	0.96		
	PVTv2-B5	0	0.97	0.99	0.98	0.97	0.96
		1	0.86	0.63	0.73		
		macro	0.92	0.81	0.86		
		weighted	0.97	0.97	0.97		
	Twins-SVT-L	0	0.99	0.97	0.98	0.96	0.98
		1	0.65	0.88	0.75		
		macro	0.82	0.93	0.86		
		weighted	0.97	0.96	0.96		
	CSWin-B	0	0.99	0.96	0.97	0.95	0.97
		1	0.59	0.87	0.70		
		macro	0.79	0.91	0.84		
		weighted	0.96	0.95	0.96		
	Swin-L	0	0.99	0.95	0.97	0.95	0.98
		1	0.56	0.93	0.70		
		macro	0.78	0.94	0.84		
		weighted	0.97	0.95	0.95		
**Diabetic** **Retinopathy** **(3-class)**	RegNetY-16GF	0	0.99	0.96	0.97	0.94	0.95
		1	0.43	0.75	0.54		0.90
		2	0.74	0.72	0.73		0.82
		macro	0.72	0.81	0.75		0.89
		weighted	0.96	0.94	0.95		0.95
	SE-ResNeXt101 -32×8d	0	0.98	0.98	0.98	0.96	0.96
		1	0.60	0.67	0.63		0.94
		2	0.81	0.64	0.72		0.98
		macro	0.80	0.77	0.78		0.96
		weighted	0.96	0.96	0.96		0.96
	PVTv2-B5	0	0.98	0.98	0.98	0.96	0.97
		1	0.65	0.65	0.65		0.96
		2	0.74	0.74	0.74		0.99
		macro	0.79	0.79	0.79		0.97
		weighted	0.96	0.96	0.96		0.97
	Twins-SVT-L	0	0.99	0.97	0.98	0.95	0.97
		1	0.58	0.71	0.64		0.96
		2	0.57	0.88	0.69		0.99
		macro	0.71	0.85	0.77		0.97
		weighted	0.96	0.95	0.96		0.97
	CSWin-B	0	0.99	0.98	0.98	0.96	0.97
		1	0.58	0.80	0.67		0.95
		2	0.81	0.76	0.78		0.99
		macro	0.79	0.84	0.81		0.97
		weighted	0.97	0.96	0.96		0.97
	Swin-L	0	0.99	0.95	0.97	0.94	0.98
		1	0.49	0.76	0.59		0.96
		2	0.56	0.89	0.69		0.99
		macro	0.68	0.87	0.75		0.98
		weighted	0.96	0.94	0.95		0.98
**Hypertension**	RegNetY-16GF	0	0.92	0.73	0.82	0.74	0.82
		1	0.42	0.75	0.54		
		macro	0.67	0.74	0.68		
		weighted	0.82	0.74	0.76		
	SE-ResNeXt101 -32×8d	0	0.90	0.80	0.85	0.77	0.83
		1	0.46	0.67	0.54		
		macro	0.68	0.73	0.69		
		weighted	0.81	0.77	0.78		
	PVTv2-B5	0	0.91	0.73	0.81	0.73	0.81
		1	0.41	0.72	0.52		
		macro	0.66	0.73	0.66		
		weighted	0.81	0.73	0.75		
	Twins-SVT-L	0	0.94	0.70	0.80	0.72	0.84
		1	0.41	0.84	0.55		
		macro	0.68	0.77	0.68		
		weighted	0.84	0.72	0.75		
	CSWin-B	0	0.92	0.76	0.83	0.75	0.83
		1	0.44	0.75	0.55		
		macro	0.68	0.75	0.69		
		weighted	0.82	0.75	0.77		
	Swin-L	0	0.91	0.81	0.85	0.78	0.85
		1	0.47	0.68	0.56		
		macro	0.69	0.74	0.71		
		weighted	0.82	0.78	0.79		
**Hypertensive** **Retinopathy**	RegNetY-16GF	0	0.99	0.93	0.96	0.92	0.67
		1	0.07	0.28	0.11		
		macro	0.53	0.61	0.54		
		weighted	0.97	0.92	0.95		
	SE-ResNeXt101 -32×8d	0	0.99	0.96	0.97	0.95	0.76
		1	0.09	0.22	0.13		
		macro	0.54	0.59	0.55		
		weighted	0.97	0.95	0.96		
	PVTv2-B5	0	0.99	0.98	0.98	0.97	0.80
		1	0.15	0.22	0.18		
		macro	0.57	0.60	0.58		
		weighted	0.97	0.97	0.97		
	Twins-SVT-L	0	0.99	0.98	0.98	0.97	0.80
		1	0.21	0.25	0.23		
		macro	0.60	0.62	0.61		
		weighted	0.97	0.97	0.97		
	CSWin-B	0	0.99	0.96	0.98	0.95	0.82
		1	0.15	0.37	0.21		
		macro	0.57	0.67	0.59		
		weighted	0.97	0.95	0.96		
	Swin-L	0	0.99	0.98	0.98	0.97	0.81
		1	0.20	0.25	0.22		
		macro	0.59	0.62	0.60		
		weighted	0.97	0.97	0.97		
**Drusen**	RegNetY-16GF	0	0.93	0.95	0.94	0.89	0.89
		1	0.72	0.65	0.68		
		macro	0.82	0.80	0.81		
		weighted	0.89	0.89	0.89		
	SE-ResNeXt101 -32×8d	0	0.93	0.93	0.93	0.88	0.88
		1	0.65	0.66	0.65		
		macro	0.79	0.79	0.79		
		weighted	0.88	0.88	0.88		
	PVTv2-B5	0	0.94	0.93	0.94	0.89	0.91
		1	0.69	0.71	0.70		
		macro	0.81	0.82	0.82		
		weighted	0.89	0.89	0.89		
	Twins-SVT-L	0	0.93	0.95	0.94	0.90	0.92
		1	0.73	0.66	0.69		
		macro	0.83	0.80	0.82		
		weighted	0.89	0.90	0.90		
	CSWin-B	0	0.95	0.92	0.93	0.89	0.92
		1	0.66	0.75	0.70		
		macro	0.80	0.83	0.82		
		weighted	0.90	0.89	0.89		
	Swin-L	0	0.94	0.94	0.94	0.90	0.93
		1	0.73	0.70	0.71		
		macro	0.83	0.82	0.83		
		weighted	0.90	0.90	0.90		
**Sex**	RegNetY-16GF	0	0.85	0.68	0.75	0.73	0.82
		1	0.61	0.80	0.69		
		macro	0.73	0.74	0.72		
		weighted	0.76	0.73	0.73		
	SE-ResNeXt101 -32×8d	0	0.79	0.79	0.79	0.74	0.80
		1	0.66	0.66	0.66		
		macro	0.73	0.73	0.73		
		weighted	0.74	0.74	0.74		
	PVTv2-B5	0	0.89	0.67	0.76	0.75	0.87
		1	0.62	0.87	0.72		
		macro	0.76	0.77	0.74		
		weighted	0.79	0.75	0.75		
	Twins-SVT-L	0	0.87	0.69	0.77	0.74	0.84
		1	0.62	0.83	0.71		
		macro	0.75	0.76	0.74		
		weighted	0.77	0.74	0.75		
	CSWin-B	0	0.84	0.82	0.83	0.80	0.87
		1	0.72	0.75	0.74		
		macro	0.78	0.79	0.78		
		weighted	0.80	0.80	0.80		
	Swin-L	0	0.85	0.83	0.84	0.80	0.87
		1	0.73	0.76	0.74		
		macro	0.79	0.79	0.79		
		weighted	0.80	0.80	0.80		

**Table 5 bioengineering-12-00840-t005:** Average inference and saliency map generation times per-image (in milliseconds) for implemented models. All experiments were conducted on an NVIDIA T4 GPU using 224 × 224 pixel images over 100 runs.

Model	Inference Time per Image (Batch = 1)	Inference Time per Image (Batch = 16)	Average Time for Saliency Map Generation per Image (Batch = 1)
**RegNetY-16GF**	13.80	9.60	50.95
**SE-ResNeXt101-32×8d**	27.58	11.93	85.47
**PVTv2-B5**	43.35	9.97	136.71
**Twins-SVT-L**	20.15	10.74	67.52
**CSWin-B**	68.56	13.56	204.82
**Swin-L**	32.91	24.41	87.83

**Table 6 bioengineering-12-00840-t006:** Performance comparison between the best-performing model in this study (Swin-L) and other investigated models on the BRSET dataset.

Label	Model	Class	Precision	Recall	F1	Accuracy	AUC
**Diabetes**	ConvNext V2 Large [26]	0	-	-	-	-	0.87
		1	-	-	-		
		macro	-	-	0.70		
		weighted	-	-	-		
	Our Study (Best Model)	0	**0.92**	**0.92**	**0.92**	**0.86**	**0.88**
		1	**0.57**	**0.57**	**0.57**		
		macro	**0.74**	**0.74**	**0.74**		
		weighted	**0.86**	**0.86**	**0.86**		
**Diabetic** **Retinopathy** **(2-class)**	ConvNext V2 Large [26]	0	-	-	-	-	0.97
		1	-	-	-		
		macro	-	-	**0.89**		
		weighted	-	-	-		
	ResNet 50 [54]	0	-	-	-	**0.97**	0.97
		1	-	-	-		
		macro	**0.86**	0.78	0.82		
		weighted	-	-	-		
	Our Study (Best Model)	0	**0.99**	**0.95**	**0.97**	0.95	**0.98**
		1	**0.56**	**0.93**	**0.70**		
		macro	0.78	**0.94**	0.84		
		weighted	**0.97**	**0.95**	**0.95**		
**Diabetic** **Retinopathy** **(3-class)**	ConvNext V2 Large [26]	0	-	-	-	-	0.97
		1	-	-	-		
		2	-	-	-		
		macro	-	-	0.82		
		weighted	-	-	-		
	ResNet-200D [55]	0	0.99	**0.99**	**0.99**	-	0.95
		1	**0.73**	0.73	**0.73**		
		2	**0.81**	0.70	**0.75**		
		macro	**0.84**	0.81	**0.83**		
		weighted	-	-	-		
	Our Study (Best Model)	0	**0.99**	0.95	0.97	**0.94**	**0.98**
		1	0.49	**0.76**	0.59		
		2	0.56	**0.89**	0.69		
		macro	0.68	**0.87**	0.75		
		weighted	**0.96**	**0.94**	**0.95**		
**Hypertension**	Our Study (Best Model)	0	**0.91**	**0.81**	**0.85**	**0.78**	**0.85**
		1	**0.47**	**0.68**	**0.56**		
		macro	**0.69**	**0.74**	**0.71**		
		weighted	**0.82**	**0.78**	**0.79**		
**Hypertensive** **Retinopathy**	ResNet 50 [54]	0	-	-	-	0.95	0.76
		1	-	-	-		
		macro	0.10	0.27	0.08		
		weighted	-	-	-		
	Our Study (Best Model)	0	**0.99**	**0.98**	**0.98**	**0.97**	**0.81**
		1	**0.20**	**0.25**	**0.22**		
		macro	**0.59**	**0.62**	**0.60**		
		weighted	**0.97**	**0.97**	**0.97**		
**Drusen**	ResNet 50 [54]	0	-	-	-	0.87	0.92
		1	-	-	-		
		macro	0.73	0.80	0.76		
		weighted	-	-	-		
	Our Study (Best Model)	0	**0.94**	**0.94**	**0.94**	**0.90**	**0.93**
		1	**0.73**	**0.70**	**0.71**		
		macro	**0.83**	**0.82**	**0.83**		
		weighted	**0.90**	**0.90**	**0.90**		
**Sex**	ConvNext V2 Large [26]	0	-	-	-	-	**0.91**
		1	-	-	-		
		macro	-	-	**0.83**		
		weighted	-	-	-		
	ResNet-200D [55]	0	-	-	-	0.75	0.80
		1	-	-	-		
		macro	-	-	-		
		weighted	-	-	-		
	Our Study (Best Model)	0	**0.85**	**0.83**	**0.84**	**0.80**	0.87
		1	**0.73**	**0.76**	**0.74**		
		macro	**0.79**	**0.79**	0.79		
		weighted	**0.80**	**0.80**	**0.80**		

## Data Availability

Access to the Brazilian Retinopathy of Prematurity Screening and Eye Tracking (BRSET) data for non-commercial research purposes can be requested through PhysioNet at https://physionet.org/content/brazilian-ophthalmological/1.0.0/ (accessed on 30 July 2025), subject to credentialed access approval.

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
