# Peer review of "Deep Learning for Comprehensive Analysis of Retinal Fundus Images: Detection of Systemic and Ocular Conditions"

_bioengineering, 2025, doi:10.3390/bioengineering12080840_

Round 1
Reviewer 1 Report
Comments and Suggestions for Authors
- The abstract should be modified to indicate the main contribution and significance on the basses of achieved results.
- A table on the hyperparameters should be included for the better understandings the adoption. The authors are encouraged to offer supporting evidence from relevant works to validate the selection and tuning of these hyperparameter.
- What type of activation function, optimizers, layer wise structure and learning rate is assumed. I suggest to read the following articles;
- SwishReLU: A Unified Approach to Activation Functions for Enhanced Deep Neural Networks Performance
- Additive Parameter for Deep Face Recognition
- Optimizing GNN Architectures Through Nonlinear Activation Functions for Potent Molecular Property Prediction.
- Figure 1 needs to be discussed in proper way.
- Overview in Table 2, needs to improve based on the resent work in this direction. As the table include the studies up to 2021, I suggest to include upto 2025.
Author Response
1 |
The abstract should be modified to indicate the main contribution and significance on the basses of achieved results. |
Thank you for your valuable suggestion. We have revised the Abstract to clearly highlight the main contribution of our work and to emphasize the significance of the achieved results. Specifically, we now report the performance of the best-performing model (Swin-L). |
2
|
A table on the hyperparameters should be included for the better understandings the adoption. The authors are encouraged to offer supporting evidence from relevant works to validate the selection and tuning of these hyperparameter. What type of activation function, optimizers, layer wise structure and learning rate is assumed. I suggest to read the following articles; SwishReLU: A Unified Approach to Activation Functions for Enhanced Deep Neural Networks Performance Additive Parameter for Deep Face Recognition Optimizing GNN Architectures Through Nonlinear Activation Functions for Potent Molecular Property Prediction. |
Thank you for the insightful suggestion. In response, we have incorporated comprehensive details regarding the selected hyperparameters, such as activation functions, optimizers, learning rates, and layer-wise architectures within the revised main text to enhance transparency and reproducibility. Additionally, a summary of all key hyperparameters and their corresponding configurations is now clearly presented in Table 3.
|
4 |
Figure 1 needs to be discussed in proper way. |
Thank you for your helpful feedback. We have revised the manuscript to provide a clear and more comprehensive explanation of Figure 1. Specifically, we now discuss the key stages of the methodology in the first paragraph of Section 2. |
5 |
Overview in Table 2, needs to improve based on the resent work in this direction. As the table include the studies up to 2021, I suggest to include upto 2025. |
Thank you for your thoughtful suggestion. In response, we have updated Table 2 with appropriate references for each model. Furthermore, to ensure the study reflects recent advancements, we evaluated several additional models beyond those presented in the main manuscript, including Mamba Vision (2025). However, as illustrated below, in the Diabetes scenario (AUC = 0.81), its performance was not competitive with our selected models. To provide a balanced and comprehensive comparison, we also assessed other well-known architectures, such as EfficientNet-B4, EfficientNet-V2L, DenseNet201, ViT-Base-Patch16, ViT-Large-Patch16, and MViTv2-Base. These models were ultimately excluded from the final table due to either: (1) noticeably lower performance in terms of AUC and F1-score, or (2) similar performance but at the cost of significantly higher computational complexity and training time. Moreover, we would like to share the fact that we tried to have a comperhensive analysis for this study, so we evaluated the performance of both ViT and CNN architectures, and this is the reason you see two CNN networks and four ViT networks. Therefore, our final model selection reflects a trade-off between performance and practicality, focusing on architectures that offer both high accuracy and computational feasibility. |

Reviewer 2 Report
Comments and Suggestions for Authors
In this article was we presented a comprehensive evaluation of six state-of-the-art deep neural networks, including convolutional neural networks and vision transformer architectures on the Brazilian Multilabel Ophthalmological Dataset, Was explored seven classification tasks, including diabetes, diabetic retinopathy (binary and three-class), hypertension, hypertensive retinopathy, drusens, and sex classification. Models were evaluated using precision, recall, F1-score, accuracy, receiver operating characteristic curve, and area under the curve. Among them, the Swin-L transformer achieved consistently excellent performance across most scenarios. Was integrated gradient-based saliency maps to enhance explainability and visualize decision-relevant retinal features.
However, there are a few things that need to be made:
- The introduction lacks reference to numerical research results.
- Please describe Figure 1 in more detail.
- For Figures 2, 3 and 4 there is no (a), (b) ....... Please complete it.
- The text lacks detailed descriptions of Figures 2, 3 and 4. Please refer to each in the text.
- The conclusions do not include numerical data on the research results.
Author Response
1 |
In this article was we presented a comprehensive evaluation of six state-of-the-art deep neural networks, including convolutional neural networks and vision transformer architectures on the Brazilian Multilabel Ophthalmological Dataset, Was explored seven classification tasks, including diabetes, diabetic retinopathy (binary and three-class), hypertension, hypertensive retinopathy, drusens, and sex classification. Models were evaluated using precision, recall, F1-score, accuracy, receiver operating characteristic curve, and area under the curve. Among them, the Swin-L transformer achieved consistently excellent performance across most scenarios. Was integrated gradient-based saliency maps to enhance explainability and visualize decision-relevant retinal features.However, there are a few things that need to be made:
The introduction lacks reference to numerical research results. |
Thank you for your valuable comment. In the Introduction section, we already cited related works along with their numerical performance results to provide context for our study. To improve clarity, we have revised and highlighted these parts accordingly. The detailed performance metrics of our proposed models are thoroughly presented and discussed in the Results section. |
2 |
Please describe Figure 1 in more detail. |
Thank you for your helpful feedback. We have revised the manuscript to provide a clear and more comprehensive explanation of Figure 1. Specifically, we now discuss the key stages of the methodology in the first paragraph of Section 2. |
3 |
For Figures 2, 3 and 4 there is no (a), (b) ....... Please complete it. |
Thank you for your observation. We have now added subfigure labels (e.g., (a), (b), etc.) to all relevant plots in Figures to ensure clarity and consistency. |
4 |
The text lacks detailed descriptions of Figures 2, 3 and 4. Please refer to each in the text. |
Thank you for pointing this out. We have revised the manuscript to include detailed descriptions and interpretations of Figures 2-6 within the main text. Each figure is now explicitly referenced and discussed to improve clarity and support the results presented. |
5 |
The conclusions do not include numerical data on the research results. |
Thank you for your helpful comment. We have revised the Conclusion section to include key numerical results from our study to strengthen the summary of our findings and their significance. |

Reviewer 3 Report
Comments and Suggestions for Authors
The manuscript’s thorough evaluation of multiple state-of-the-art deep learning models on a large multilabel fundus image dataset is commended. However, several methodological refinements are warranted.
Major points
- The manuscript largely rehashes existing architectures on a new dataset without offering unique conceptual contributions. To resolve this, the authors should articulate and validate at least one novel insight – such as the first demonstration of systemic condition detection (e.g., hypertension) or sex classification from fundus images – and ideally design targeted experiments or analyses (e.g., ablation studies) that highlight how their findings advance beyond prior work.
- Critical hyperparameters and preprocessing steps are omitted, undermining reproducibility. The authors should fully disclose optimizer choices, learning rates and schedules, batch sizes, number of epochs, weight decay, data augmentation pipelines (including resizing, cropping, color transformations), and whether they fine-tuned all layers or only classifier heads, along with random seeds and software frameworks, so that peers can replicate and extend their work.
- Reported metrics lack confidence intervals or hypothesis testing to establish significance between models. The authors should compute and present 95% confidence intervals (e.g., via bootstrapping) around AUC and F1 scores, perform paired statistical tests such as DeLong’s test for AUC differences, and include calibration measures (e.g., Brier score, reliability diagrams) to demonstrate both discrimination and calibration for clinical reliability.
- Gradient-based saliency maps are used without quantitative validation or comparison to more robust techniques. To strengthen interpretability claims, the authors should include at least one additional method (e.g., Grad-CAM, integrated gradients, SHAP), perform a quantitative “pointing game” or overlap analysis against expert-annotated lesion masks, and solicit clinician feedback to confirm that highlighted regions correspond to true pathology.
Minor points
- Performance on rare classes like hypertensive retinopathy is likely inflated by the severe imbalance in BRSET. The authors must implement and compare imbalance mitigation strategies (e.g., focal loss, oversampling of minority classes, synthetic augmentation), report their effects on recall and precision for these classes, and include precision–recall curves alongside ROC curves to provide a realistic appraisal.

Author Response
|
The manuscript’s thorough evaluation of multiple state-of-the-art deep learning models on a large multilabel fundus image dataset is commended. However, several methodological refinements are warranted.
Major points The manuscript largely rehashes existing architectures on a new dataset without offering unique conceptual contributions. To resolve this, the authors should articulate and validate at least one novel insight – such as the first demonstration of systemic condition detection (e.g., hypertension) or sex classification from fundus images – and ideally design targeted experiments or analyses (e.g., ablation studies) that highlight how their findings advance beyond prior work. |
Thank you for your insightful feedback. We respectfully clarify that our study presents several novel contributions, particularly in evaluating systemic condition detection, such as hypertension from retinal fundus images using state-of-the-art ViT architectures. These scenarios are explicitly included among our seven classification tasks. To the best of our knowledge, this is the first study to benchmark hypertension detection on the BRSET dataset using modern ViTs, thereby establishing a new performance baseline for this underexplored application. Although sex classification from fundus images has been explored previously, we included this task to comprehensively assess model performance across both ocular and systemic features. Our results show that our Swin-L model outperforms prior reported metrics, underscoring the effectiveness of our multi-label framework and the advantages of advanced transformer-based architectures. In addition to model selection and task diversity, our study emphasizes methodological transparency and rigor. For each classification scenario, we report multiple performance metrics, including accuracy, precision, recall, F1-score, ROC curves, AUC, and PR curves. We provide results in macro-averaged, weighted, and per-class. This level of detailed reporting exceeds that of most prior studies and offers a more comprehensive evaluation of model performance, particularly in the context of class imbalance and clinical interpretability. We have revised the manuscript to better emphasize these contributions and to more clearly articulate how our findings advance the current state of knowledge in retinal image analysis.
|
|
Critical hyperparameters and preprocessing steps are omitted, undermining reproducibility. The authors should fully disclose optimizer choices, learning rates and schedules, batch sizes, number of epochs, weight decay, data augmentation pipelines (including resizing, cropping, color transformations), and whether they fine-tuned all layers or only classifier heads, along with random seeds and software frameworks, so that peers can replicate and extend their work.
|
Thank you for the insightful suggestion. In response, we have incorporated comprehensive details regarding the selected hyperparameters, such as activation functions, optimizers, learning rates, and layer-wise architectures within the revised main text to enhance transparency and reproducibility. Additionally, a summary of all key hyperparameters and their corresponding configurations is now clearly presented in Table 3. |
|
Reported metrics lack confidence intervals or hypothesis testing to establish significance between models. The authors should compute and present 95% confidence intervals (e.g., via bootstrapping) around AUC and F1 scores, perform paired statistical tests such as DeLong’s test for AUC differences, and include calibration measures (e.g., Brier score, reliability diagrams) to demonstrate both discrimination and calibration for clinical reliability. |
Thanks for your valuable suggestion. However, we respectfully clarify that such analyses fall outside the primary scope and objective of this study. Our work focuses on a comprehensive benchmarking of six modern DNN architectures, spanning both CNN and ViT designs, across seven clinically relevant classification tasks using the BRSET dataset. The study was designed to explore comparative performance across architectures, using consistent and widely adopted evaluation metrics such as AUC, F1-score, accuracy, precision, and recall, supported by five-fold cross-validation to ensure robustness. The inclusion of confidence intervals or hypothesis testing, while informative in pairwise model comparisons, would be more appropriate in studies specifically aimed at model ranking, statistical model selection, or regulatory validation, especially when models are already close in performance and decisions must be made about clinical deployment. Our primary goal, in contrast, was to explore diagnostic feasibility, establish performance baselines, and demonstrate the viability of retinal images for both ocular and systemic disease classification using modern transformer-based models. Similarly, while calibration metrics (e.g., Brier score or reliability diagrams) are crucial in risk prediction models or probabilistic inference scenarios, our current work is oriented toward classification tasks where model discrimination (as captured by AUC and F1-score) is of primary concern. Moreover, the BRSET dataset lacks temporally or geographically external validation cohorts, which are generally required for meaningful calibration analysis in clinical applications. |
|
Gradient-based saliency maps are used without quantitative validation or comparison to more robust techniques. To strengthen interpretability claims, the authors should include at least one additional method (e.g., Grad-CAM, integrated gradients, SHAP), perform a quantitative “pointing game” or overlap analysis against expert-annotated lesion masks, and solicit clinician feedback to confirm that highlighted regions correspond to true pathology. |
Thank you for your thoughtful comment. In response, we have extended our interpretability analysis by incorporating two additional gradient-based saliency methods: Integrated Gradients and GradientSHAP, alongside the previously used Vanilla Gradients. These additions strengthen our interpretability claims by providing diverse perspectives on the model's decision-making process. However, we would like to clarify a key information of the dataset. The BRSET dataset provides only image-level labels for classification tasks (e.g., presence or absence of a condition) and does not include pixel-level annotations or lesion masks that would enable quantitative validation techniques such as the “pointing game” or overlap analysis. Consequently, methods that rely on object localization or segmentation ground truth—such as comparison to expert-annotated pathology boundaries are not feasible in this context. As such, our focus has been on qualitative interpretability using saliency maps to highlight decision-relevant regions. While we acknowledge the value of clinician feedback, the current dataset does not support spatial alignment with clinical annotations. We have added this clarification to the manuscript to transparently communicate the limitations and rationale for our chosen approach. |
|
Performance on rare classes like hypertensive retinopathy is likely inflated by the severe imbalance in BRSET. The authors must implement and compare imbalance mitigation strategies (e.g., focal loss, oversampling of minority classes, synthetic augmentation), report their effects on recall and precision for these classes, and include precision–recall curves alongside ROC curves to provide a realistic appraisal. |
Thank you for your constructive suggestion. We fully acknowledge the challenge posed by severe class imbalance in the BRSET dataset, particularly for rare conditions such as hypertensive retinopathy. To promote transparency, we reported all evaluation metricsÙˆ including per-class, macro-averaged, and weighted scores—as shown in the main results tables. We made another effort to address class imbalance as we implemented and tested several mitigation strategies. Specifically, we used Weighted Cross-Entropy Loss, which assigns greater penalty to minority classes and is widely adopted for handling imbalance in deep learning [Buda et al., 2018]. Additionally, we experimented with Focal Loss using the Swin-L model for the hypertensive retinopathy scenario. However, focal loss led to degraded performance: AUC dropped from 0.81 to 0.79, accuracy decreased from 0.97 to 0.93, and macro F1-score declined from 0.60 to 0.56. These results are shown in the figure below. Based on these findings—and in consultation with our expert ophthalmologist—we concluded that overly aggressive imbalance correction may distort the clinical interpretation of model performance. Therefore, we reported results using Weighted Cross-Entropy Loss, which yielded better and more stable outcomes. We also revised the manuscript to explicitly include precision–recall (PR) curves for all classification tasks involving rare classes, as shown in Figures 4 and 5, in addition to the ROC curves. This provides a more realistic appraisal of model performance under class imbalance. Reference: Buda, M., Maki, A., & Mazurowski, M. A. (2018). A systematic study of the class imbalance problem in convolutional neural networks. Neural Networks, 106, 249–259. |

Round 2
Reviewer 3 Report
Comments and Suggestions for Authors
The corrections made the paper much better and now it can be reomended for publication in the present form.